# Purification and Characterization of Authentic 30S Ribosomal Precursors Induced by Heat Shock

**DOI:** 10.3390/ijms24043491

**Published:** 2023-02-09

**Authors:** Emmanuel Giudice, Sylvie Georgeault, Régis Lavigne, Charles Pineau, Annie Trautwetter, Gwennola Ermel, Carlos Blanco, Reynald Gillet

**Affiliations:** 1Univ Rennes, CNRS, Institut de Génétique et Développement de Rennes (IGDR) UMR6290, 35000 Rennes, France; 2Univ Rennes, Inserm, EHESP, Irset (Institut de Recherche en Santé, Environnement et Travail)-UMR_S 1085, 35000 Rennes, France; 3Univ Rennes, CNRS, Inserm, Biosit UAR 3480 US_S 018, Protim Core Facility, 35000 Rennes, France

**Keywords:** 21S, DnaK, heat shock, ribosome assembly, cryo-EM

## Abstract

Ribosome biogenesis is a complex and multistep process that depends on various assembly factors. To understand this process and identify the ribosome assembly intermediates, most studies have set out to delete or deplete these assembly factors. Instead, we took advantage of the impact of heat stress (45 °C) on the late stages of the biogenesis of the 30S ribosomal subunit to explore authentic precursors. Under these conditions, reduced levels of the DnaK chaperone proteins devoted to ribosome assembly lead to the transient accumulation of 21S ribosomal particles, which are 30S precursors. We constructed strains with different affinity tags on one early and one late 30S ribosomal protein and purified the 21S particles that form under heat shock. A combination of relative quantification using mass spectrometry-based proteomics and cryo-electron microscopy (cryo-EM) was then used to determine their protein contents and structures.

## 1. Introduction

Bacteria have developed a myriad of mechanisms to allow them to grow in hostile environments or under stressful conditions. Because exposure to elevated temperatures causes a constant denaturation of proteins, heat stress has a large impact on bacterial fitness and induces a regulated response. The molecular mechanisms that control this specific cellular stress response are mainly governed at the transcriptional level by the heat-shock sigma factor RpoH. The activation of RpoH at high temperatures enables the increased expression of heat shock proteins (HSPs). Many HSPs function as chaperones, which are needed for both the correct folding of neosynthesized proteins and for the renaturation or degradation of denatured proteins [1]. While these proteins play essential roles in cellular metabolism under regular conditions, their synthesis increases dramatically when temperatures are elevated. 

Among the many macromolecular complexes affected by stress, ribosomes are of major importance. Indeed, their activity is crucial for cell viability, and they can sense the cell’s metabolic state through their interactions with cofactors, notably the Rel/RelA enzymes which monitor the aminoacylation status of ribosomal A-site tRNA [2]. In fact, these enzymes are positively regulated by their product, the alarmone nucleotide (p)ppGpp, which is an activator of the "stringent" response which in turn affects ribosome synthesis [3,4,5]. These various mechanisms are very well described, but ribosome biogenesis under stressful conditions is less understood. 

Ribosome synthesis begins with the transcription of a large ribosomal RNA (rRNA) precursor molecule that is then cleaved by RNase III into the precursors of the three rRNAs. The assembly process is sequential and cooperative, with a series of RNA processing and conformational changes as well as the hierarchical binding of ribosomal proteins [6,7]. These events are favored by the intervention of many ribosome assembly factors that regulate the assembly of the large (50S) and small (30S) ribosomal subunits [6,8].

The 30S late-assembly intermediates that accumulate in *E. coli* null mutants deleted from the assembly factors *rimM, rbfA, yjeQ* and *era* have already been isolated and well-described [9,10,11,12,13,14,15]. These immature precursors are structurally related, all resembling the mature 30S subunit but with severe distortions at the decoding center which make translation impossible [16]. Each of these precursors are blocked in the later stages of ribosome maturation, which suggests that multiple parallel assembly pathways exist, and that these pathways converge at the end of the assembly process [17]. In the same way, severe heat stress also affects ribosome biogenesis, inducing the accumulation of ribosome precursors. The formation of these precursors after chaperone titration by denatured proteins has been observed both in vitro and in vivo. Indeed, the depletion of DnaK or GroEL chaperones causes the immature ribosomes to accumulate, indicating that these chaperones are most probably involved in ribosome maturation [18,19]. Accordingly, at temperatures below 30 °C, *dnaK* and *dnaJ* mutants exhibit less growth than the wild type, but with no ribosome assembly defects. At temperatures above 42 °C, however, ribosome assembly decreases, while both 30S and 50S precursors accumulate [20]. The overproduction of GroEL chaperones can suppress defects in the large 50S subunit assembly, but does not affect the maturation of the 30S precursor [21]. A transient accumulation of this precursor, the 21S particle, was also observed in wild-type strains subjected to severe heat stress [22]. These are genuine precursors that slowly convert into 30S mature subunits, but are produced in very small amounts. Until now, this had unfortunately prevented their isolation as well as the analysis of their composition and structure, thus limiting our understanding of the influence of heat stress on ribosome maturation at the molecular level. 

In this study, we constructed bacterial strains with affinity tags on early and late 30S ribosomal proteins, which enabled us to purify 21S particles that had been formed under severe heat shock conditions. We determined their protein contents and structures using a combination of relative quantification using mass spectrometry-based proteomics and cryo-electron microscopy (cryo-EM).

## 2. Results

### 2.1. Introduction of His and Strep Affinity Tags onto 30S Ribosomal Proteins

Because of the reduced amount of DnaK chaperones devoted to ribosome assembly under heat shock conditions, there is a transient accumulation of authentic precursors, not degraded or dead-end particles. These 21S particles were previously characterized after radioactive labelling, since very few are normally present in bacterial cells [22]. The isolation of these small ribosomal subunits therefore depends on having a very efficient method for eliminating mature particles, and affinity chromatography is an attractive solution for the purification of 21S. We noted that the 30S precursors described in the literature contain bS20 but not uS2 [9,10,11,12,13,17]. We therefore decided to introduce two different tags, using strep tags on bS20, and 6xHis tags on the late-assembly uS2 proteins [23]. We chose bS20 for the early proteins because its N- and C-terminal extremities are accessible enough for the tag to be used to pull out the particles, yet its core is deeply buried, which prevents dropping during the process. We used the Datsenko and Wanner method [24] to insert these tags at the *rpsB* (encoding the late protein uS2) and *rpsT* (encoding the early protein bS20) carboxyl termini in E6006 bacteria. Because the disruption of ribosomal biogenesis or maturation can cause slower growth or cold sensitivity, we first checked whether the tag insertions affected bacterial growth. To do this, we compared the growth characteristics of E6006 to those of MG1655 at 20, 30, 37 and 42 °C in LB and M63 minimal medium. In all conditions, the growth was identical in both strains. Similarly, we found that the presence of a tag did not affect the minimum inhibitory concentrations (MICs) for gentamycin, kanamycin, chloramphenicol, or in vitro translation assays (results not shown). These data suggest that tag presence does not affect ribosome activity. By incorporating these tags, we were able to purify ribosomes very efficiently using nickel-charged affinity resin (Ni-NTa) or streptactin. Finally, we used the E6006 strain to purify 70S from crude extracts, as well as for obtaining 30S and 50S ribosomes from ribosome pellets obtained from a single ultracentrifugation (Figure 1). 

### 2.2. Ribosome Synthesis at 45 °C

Before purification of the 21S particles, we checked to ensure that 70S ribosomes could be correctly neosynthesized at this high temperature. Since bS20 proteins repress their own synthesis [25], we cloned an *rpsT*-strep open reading frame deprived of regulatory sequences into the pBAD24 vector under the control of the p*araB* promoter. This ensured that bS20-strep would not repress its own synthesis but would instead repress the chromosomally encoded bS20. We grew the cells at 30 °C in LB medium containing glucose to block bS20:strep synthesis, then collected them by centrifugation and re-suspended them in LB medium. We divided the cells into two groups and cultivated them at either 30 or 45 °C until an OD_600_ of 0.8. We induced the production of bS20:strep by adding 1 mM L-arabinose. We then collected and broke the cells, and the crude extracts were applied to streptactin columns (see Section 4 Materials and Methods). The eluted fractions of both groups (induced at 30 or 45 °C) show similar profiles (Figure 1C, lane 4), revealing the presence of 70S ribosomes. We therefore showed that ribosome synthesis is not compromised, and that mature ribosomes can form during heat stress of 45 °C. This is not surprising, since ribosome synthesis is mostly dependent on growth rates, and these are almost identical under both temperature conditions [26].

### 2.3. Isolation and Affinity Purification of Tagged 21S Intermediates

We submitted the E6006 strain to a heat shock of 45 °C for 60 min. We then isolated the ribosomes by ultracentrifugation on a 10–40% sucrose gradient. To pinpoint the location of 21S ribosomal particles, we performed hybridization of the appropriate biotinylated oligonucleotides as well as RNA gel electrophoresis (Figure 2). The oligonucleotides targeting the non-mature rRNAs (21S-115 and 21S-33) hybridize to the RNAs from fractions 10 to 19 (Figure 2C), and a supplementary band above the 16S rRNA is recovered in fractions 10 to 16 (Figure 2D). This reveals the presence of 17S rRNA, a pre-maturation form of 16S rRNA [22], in the first peak (fractions 10 to 22, Figure 2A) of the ribosome profile. 

The 17S precursor rRNA suggests that 21S early ribosomal particles must be present just before the 30S peak. We therefore collected these fractions containing 21S and discarded the others. We eliminated the remaining 30S mature particles by three successive nickel affinity chromatography passes. The resulting fractions contained the expected ribosomal proteins as well as various contaminating proteins, but these were easily eliminated by purifying the 21S fractions on streptactin (Figure 3).

Using the A260nm absorbance data of the various fractions, we estimated the 21S ribosomal particles to be about 1% of the crude extract total rRNA. We next purified 200 ± 50 pmoles of 21S from a total volume of 30 L of cultures after heat shock. The same protocol was applied to E6006 cells grown at 30 °C (without heat shock), but no 21S particles were detected. The 21S particles were also absent in cells left to recover at 30 °C for 1 h after heat shock. These results are in accordance with the previous observation [22] that ^3^H 17S rRNA readily converts to 16S when cells are incubated at 30 °C after undergoing heat stress.

### 2.4. Composition of 21S Proteins

As expected, an analysis of 21S particles by SDS-PAGE showed that nearly all contaminants were eliminated and that uS2 was absent (Figure 3). In the 21S fraction, we did not see the ‘atypical’ ribosomal protein bS1, weakly associated with the 30S subunit [28], and the levels of uS3, uS5, and uS7 were reduced. In order to compare 21S protein composition with that of 30S mature subunits, we then used relative quantification using mass spectrometry-based proteomics analysis (Appendix A). By comparing the protein ratios (21S/30S), we were able to define three groups based on their occupancies [29], (Figure 4A). The first set of proteins contains bS20 (which was an expected result, since it was used to fish out the particles), bS6, uS8, uS12, and uS15, with all of these found at just about equal stoichiometry as that of mature 30S. The second group contains uS4, uS7, uS9, uS11, uS14, bS16, uS17, and uS19 proteins at levels varying from 0.6 to 0.8 compared to that of the control. The third and most abundant group contains proteins bS1, uS2, bS21, uS3, uS10, bS18, uS5, and uS13, but in even lower ratios (0.1 to 0.5). Of these proteins, bS1 and uS2 are almost entirely absent, and since the screening process had previously removed all uS2-containing particles, the tiny amounts of uS2 seen must reflect a slight contamination of 21S by 30S particles. Although the protein contents of 30S precursors differ in each study previously published, some similarities have been observed. For instance, bS1 is always absent, surely because of its weak interaction with 30S. However, the native 30S we obtained here contain similar amounts of bS1 and bS20. This suggests that (under our purification conditions) bS1 is not released by the 30S subunits, and that the previously observed absence of bS1 in 21S could be due to it not being incorporated at an early stage of the small subunit biogenesis. uS3 is also under-represented, as observed in most published reports on 30S precursors [9,12,16,17,30]. In fact, almost all of the proteins found in low abundance are secondary (S9, S13, and S19) or tertiary (S10, S14, S3, and S2) binding proteins that bind to the 21S complex later in the 30S maturation process. This reflects the fact that protein binding dynamics remain high throughout the formation of the final 30S subunit, and we confirmed this by undertaking cryo-EM studies of the 21S particles (see below).

### 2.5. Determination of the Structure of 21S

We characterized the structure of purified 21S particles with cryo-EM. After semi-manual picking with CryoSPARC, we selected 172,000 particles. Following 3D classification, we retained a main class of 49,000 particles, with 28% of the particles classified (Appendix A). This allowed us to produce a final 3D reconstruction at a resolution of 6.1 Å (Figure 5A), which was detailed enough for us to describe large conformational changes and explore protein occupancy levels. The 21S head is slightly more rotated than in a mature 30S (Figure 5B,C). Contrary to other immature 30S subunits that accumulate in Δ*rimM* cells [12], the entire helix 44 is clearly visible in the 21S, with its upper region latched onto the decoding center (Figure 5D). To estimate the protein occupancy of our 21S particles, we compared our reconstruction with a model of the *E. coli* 30S ribosomal subunit (PDB 4V4Q) filtered at the same resolution. Proteins uS2, bS21, and bS1 are clearly missing from our 21S reconstruction, and the uS7 protein is very poorly defined (Figure 6A). For the most part, these observations match the occupancy levels observed using mass spectrometry. Indeed, the 21S missing proteins uS2, bS21, and bS1 are the least present in the proteomics quantification experiments (Figure 4A, group 3). The notable exception is uS7, which is relatively abundant in our quantification results (Figure 4A, group 2), but whose low cryo-EM density suggests a partial occupancy in 21S. This primary protein binds early to the 3′ domain of the 16S rRNA to initiate the 30S assembly, resulting in the fixation of seven other ribosomal proteins (Figure 4B). While the lack of uS7 and not its related secondary proteins in our 21S structure would seem to contradict a strictly sequential and cooperative ribosome assembly scheme, it was recently shown that uS7 binding dynamics during assembly gradually decrease as additional components are installed [31]. The results from Duss et al. suggest a different and more flexible process than previously thought, with the later-binding ribosomal proteins also chaperoning the rRNA folding early in assembly, as well as regulating the dynamics of protein-RNA binding during the 30S assembly. 

The differences observed could therefore be explained by the different methods employed [29]. Indeed, MS analysis reflects the average protein occupancy of the entire population, while cryo-EM structures are created from individual populations selected in silico during the single-particle refinement process. In addition, some proteins (such as uS7) may appear in the cryo-EM map because of their low flexibility or high affinity. The two methods are, however, complimentary, and their combination provides both general and individual details about the particles being observed. In fact, our structure’s global conformational changes and missing ribosomal proteins are very similar to what happens when mature 30S subunits encounter RsgA (also known as YjeQ). Although this ribosome assembly factor was absent in our reconstruction (Figure 6B), it intervenes during the later stages of 30S maturation [32,33], along with Era, RbfA, and KsgA. RsgA binds to immature 30S after RbfA, and this leads to the folding of the upper region of helix 44 and the release of RbfA [34]. Accordingly, RsgA’s binding affinity to these immature particles is weak, suggesting that they transition to a more thermodynamically stable assembly intermediate conformation which is no longer recognized by RsgA [16]. We are therefore confident that the 21S population which we analyzed represents a later stage after RsgA release.

## 3. Discussion

Until now, in vivo characterization of ribosome precursors has mainly been done by deleting or depleting ribosome assembly factors to force the accumulation of precursor particles. However, such mutations dramatically affect cellular metabolism and growth, resulting in significant feedback regulations which then affect ribosome synthesis. Here we took advantage of the increase in precursor content which occurs when cells are affected by heat shock. Indeed, the DnaK molecular chaperone induced upon heat shock to promote ATP-dependent refolding or degradation of damaged proteins is also necessary for the later steps of ribosome assembly. Under heat shock, its quantity is limited, since it is titrated by unfolded proteins, and this results in a transient reduction in the amounts of this chaperone that can be devoted to ribosome assembly, resulting in the temporary accumulation of ribosomal precursors [22]. However, this does not compromise ribosome synthesis because these precursors slowly mature into 30S ribosomal subunits, suggesting that they are authentic precursors. Indeed, similar amounts of ribosomes are neosynthesized at both 30 and 42 °C, which is unsurprising since the growth rates at these two non-optimal growth temperatures are nearly identical. We constructed strains with tags on certain ribosomal proteins. These chromosomal tags had no effect on growth at any temperature, nor did they affect antibiotic susceptibility or in vitro translation, so these tags do not affect bacterial metabolism or ribosomal activity. Even if we cannot exclude the idea that heat stress could affect ribosome biogenesis by changing the transcription of rRNA and r-protein operons, the effect of heat stress on transcription has already been analyzed in many studies, which have concluded that rRNA operon transcription is not affected at 45 °C [35,36,37,38]. We analyzed ribosome biogenesis data (Appendix A) from a transcriptomic study of the effects of heat stress on the MG1655 strain grown in the same medium and under identical heat constraints as studied here [37]. There, the expression of ribosomal protein genes was reduced immediately after stress but was maintained at high levels, which is in accordance with the reduced growth rates known to occur after a shift to high temperatures [39]. This is particularly true for proteins that are poorly represented in the 21S subunit, whose reduction in 21S is not correlated with a similar reduction in their synthesis. These data suggest that the synthesis of ribosomal components is not dramatically affected by growth at very high temperatures, so the accumulation of precursors is instead due to a defect in ribosome assembly. When cells are grown at 45 °C, expressions of the KsgA, RimM, RsgA, RbfA, and RimP maturation factors follow the same pattern as that of ribosomal proteins: there is an immediate decrease after stress, but their expressions remain high (Appendix A). This is not surprising, since many of these genes are part of operons that also encode for ribosomal proteins. This suggests that the moderate reduction of these ribosome assembly factors does not cause any drastic effects. 

From our data, we saw structural defects in the head of 21S particles. This would make these precursors incapable of translation, particularly since bS21 is absent, as that protein is necessary for proper translation initiation [40]. Actually, our 21S particles are very similar in protein content and structure to the precursors accumulated in *rim, ksgA,* and *yjeQ* mutants [9,10,11,12,13], and to those appearing after RsgA is added to mature 30S particles [33]. This suggests that heat stress and the subsequent reassignment of DnaK to denatured proteins affects the late stages of maturation, and/or the synthesis of these maturation factors. In fact, the mechanism used by the DnaK chaperone for 30S assembly is not very well understood. In vitro, DnaK facilitates 30S subunit assembly [18,19], but in vivo, it is necessary for ribosome assembly only when temperatures are above 42 °C [21,41]. Therefore, chaperones might encourage the incorporation of ribosomal proteins by various mechanisms, participating in ribosomal protein folding, assisting in their direct incorporation into the ribosome, or affecting assembly factors. In bacteria, chaperones have an important role in regulating the response to heat shock. The GroE chaperonin machine modulates the activity of the HrcA repressor in both *Bacillus subtilis* [42] and *Chlamydia trachomatis* [43]. Similarly, DnaK stimulates the operator-binding activity of the HspR protein in mycobacteria [44]. In *E. coli*, DnaK also sequestrates sigma32 by direct association, and then mediates its degradation by proteases [45]. Upon heat stress, DnaK binds preferentially with denatured proteins, so it interacts poorly with the stress regulators which allow for heat stress protein synthesis. When cells return to an optimal growth temperature, the concentration of denatured proteins decreases and free DnaK interacts with stress regulators, thereby halting the transcription of the HSP gene. Both DnaK and GroE protect proteins against denaturation, but they may also participate in their incorporation, which is consistent with their in vitro role in ribosome assembly [18,19]. Alternatively, chaperones could also protect maturation factors or amplify their activity. The fact that all intermediates are very similar strengthens the idea that there is a major step in ribosome maturation which is thermodynamically difficult, and which involves various maturation factors.

## 4. Materials and Methods

### 4.1. Bacterial Strains, Growth Media, and Genetic Procedures

The *Escherichia coli* strains and plasmids used are listed in Table 1. Bacteria were grown aerobically in LB or M63 medium [46] at 20 to 45 °C. Ampicillin and kanamycin were used at 50 µg/mL. Standard procedures were used for generalized transduction with P1, transformation, PCR, and western blotting [47]. Table 2 lists the oligonucleotides used.

### 4.2. Tagging of uS2 and bS20

The primers rpsTREPF-rpsTREPR (for *rpsT)* and rpsBHISF-rpsFR (for *rpsB)* were used to amplify the KanR gene from pKD4 [24]. The PCR products were then recombined into the *E. coli* MG1655 chromosomes using their protocol. MG1655 carrying the pKD46 plasmid was grown in LB medium with ampicillin and L-arabinose at 30 °C to an approximate OD_600_ of 0.6, then made electrocompetent [47]. PCR products were gel-purified, digested with *Dpn*I, re-purified, and then suspended in 10 mM Tris (pH 8.0). Electroporation was performed according to the manufacturer’s instructions in a BIO-RAD Gene Pulser II (2.5 kV, 25 µF, 200 Ω) with 0.25-cm chambers. Transformants were selected on LB-kanamycin plates at 42 °C. To check for possible loss of the pKD46 plasmid, the colonies were tested for ampicillin sensitivity. The chromosomal structure of mutants was confirmed by PCR (Appendix A). KanR colonies were then transformed with pCP20 plasmid. The resulting AmpR transformants were selected at 30 °C, incubated at 42 °C, and tested for antibiotic resistance. A PCR was also used to confirm the mutations of *rpsB* and *rpsT* (Appendix A). The *rpsB*:his and *rpsT*:strep strains were named ‘E6001’ and ‘E6004’, respectively. The *rpsB*-KanR mutation was transduced into E6004. Transductants were selected as KanR colonies, and then pCP20 was introduced by transformation. The KanR-AmpR colonies were isolated and their chromosomal structures were confirmed by PCR. Finally, KanR was deleted after induction of Flp recombinase, resulting in an ‘E6006’ strain containing both *rpsT*:his and *rpsB*:strep.

### 4.3. Cloning of rpsT:Strep

The *rpsT:strep* gene was amplified from the chromosomal DNA of E6004 using rpsTA and rpstR primers (Table 2). To produce pE7867, amplified DNA was cleaved with *Nco*I (at rpsTA) and with *Xba*I (in the FRT sequence), then ligated with pBAD24 cleaved with the same enzymes.

### 4.4. Purification of 30S Precursors

E6006 bacteria were grown at 30 °C in M63 medium [46] with 10 mM glucose to an OD_600_ of 0.2. The cells were then heated to 45 °C by adding medium that had been pre-warmed to 60 °C. The cultures were incubated for 1 h, then cooled to 4 °C on ice. Cells were collected by centrifugation at 8000× *g* for 10 min, with the resulting pellets re-suspended in buffer A (50 mM Tris-HCl pH 7.5, 15 mM Mg(C_2_H_3_O_2_)_2_, 100 mM NH_4_Cl, 3 mM HOCH_2_CH_2_SH, 0.5 mM EDTA, and 1 mM PMSF) and disrupted using a French press (1250 PSI). The lysate was centrifuged at 15,000× *g* for 30 min at 4 °C. The supernatant (250 A_260_ units) was centrifuged on a 38-mL (10–40%) sucrose gradient using an SW 28 rotor working at 21,000 rpm for 17 h at 4 °C. Fractions of interest were then pooled from different gradient tubes and concentrated via centrifugation at 100,000× *g* for 6 h. Pellets were dissolved in buffer A then analyzed using IMAC chromatography with Ni-NTA resin (Qiagen, France). All 70S ribosomes and 30S fractions containing his-tagged uS2 proteins were eliminated by three successive Ni-NTA purification steps. Particles retained by the resin were eluted with 100 mM imidazole in buffer A. The effluent underwent chromatography on Strep-Tactin sepharose (IBA, Göttingen, Germany) to bind particles containing the bS20 strep protein. These particles were eluted with 2.5 mM desthiobiotin in buffer A, concentrated with Amicon (100MCW) cartridges, and then analyzed by SDS-PAGE electrophoresis. The purification of mature 30S subunits from E6006 cells was done in the same way as described here for the precursors.

### 4.5. RNA Experiments

For dot blots and gel electrophoresis, total RNAs from the desired sucrose gradient fractions (500 µL) were isolated by acid guanidinium thiocyanate-phenol-chloroform (AFPC) extraction using Invitrogen TRIzol reagent (Thermo Fisher, Waltham, MA, USA). The RNA concentrations were determined using spectrophotometry (for single-stranded RNA, one OD_260_ corresponds to 40 µg/mL). RNA electrophoresis was performed using MOPS buffer (40 mM MOPS pH7, 10 mM Na acetate, and 1 mM Na EDTA), and the 2% agarose gel was loaded with 2 µg of each fraction’s RNA. For dot blots, 2 µg of RNAs were applied on Amersham Hybond-N + nylon membranes (GE Healthcare Life Sciences, Singapore) using a dot blot apparatus. After UV-fixation of the RNAs, the membrane was hybridized with the corresponding biotinylated oligonucleotides using reagents from a North2South kit (Thermo Fisher). Chemiluminescence was recorded using a ChemiDoc XRS+ system (Biorad, Hercules, CA, USA).

### 4.6. Tandem Mass Spectrometry (MS/MS) Analysis

The concentrations of 21S purified particles were normalized according to their OD_260_ values, and then mixed with a standardized amount of β-galactosidase. MS/MS analysis was undertaken by the Proteomics Core Facility of the University of Rennes 1. The Mascot search engine (Matrix Science, Boston, MA, USA) was used for protein identification, and spectral counts were calculated using ProteoIQ (Premier Biosoft). The spectral counts for each protein in a sample were normalized to that of β-galactosidase. Since 21S was purified using the bS20 Strep-tag, the relative levels of ribosomal proteins associated with 21S particles are reported here as percentages of bS20. Results were obtained from three independent cultures and affinity purifications, with two technical replicates performed for each sample.

### 4.7. Protein Identification Using Liquid Chromatography with Tandem Mass Spectrometry (LC-MS/MS)

#### 4.7.1. Liquid Digestion of Ribosomal Samples

The digestion protocol was begun by adding 53.5 μL ammonium bicarbonate buffer (50 mM, pH 7.8) to 25 µg ribosomes and ribosomal subunits, resulting in a total volume of 65 μL. After mixing for 1 min, tubes were incubated for 10 min at 80 °C, then sonicated for two minutes. The disulfide bond reduction was conducted by adding 12.5 μL DTT (65 mM) to the samples, agitating for one minute, then incubating them for 15 min at 37 °C. Reduced disulfide bonds were then alkylated by the addition of 135 mM iodoacetamide. The samples were then incubated in the dark for 15 min under agitation at room temperature. Finally, the proteins were digested overnight with 10 μL of modified 0.1 μg/μL Trypsin (Promega, Madison, WI, USA) in 50 mM of ammonium bicarbonate buffer.

#### 4.7.2. LC-MS/MS Analysis

The MS shotgun analysis was done with an UltiMate 3000 (Thermo Fisher Scientific, Bremen, Germany) nanoflow high-performance liquid chromatography (HPLC) system connected to a hybrid LTQ-Orbitrap XL (Thermo Fisher Scientific, Bremen, Germany) equipped with a nanoelectrospray ion source (New Objective). This HPLC system included a solvent degasser, a nanoflow pump, a thermostatted column compartment kept at 30 °C, and a thermostatted autosampler kept at 8 °C to reduce sample evaporation. This system uses two mobile phases: mobile A (99.9% Milli-Q water and 0.1% v/v formic acid); and mobile B (99.9% acetonitrile and 0.1% v/v formic acid). An aliquot of 10 μL of prepared peptide mixture was loaded over 3 min at a flow rate of 25 μL/min onto a PepMap C18 pre-column trap (5 mm × 300 μm i.d., 300 Å pore size, 5 μm, Thermo Fisher Scientific, Bremen, Germany) in 2% buffer B. This step was followed by reverse-phase separations (0.250 μL/min) using a PepMap C18 analytical column (15 cm × 300 μm i.d., 300 Å pore size, 5 μm). We ran a gradient from 2–35% buffer B for the first 60 min, with 35–60% buffer B during minutes 60 to 85, and finally with 60–90% buffer B for minutes 85 through 105. The column was washed with 90% buffer B for 16 min and then with 2% buffer B for 19 min, after which the next sample was loaded. The peptides were detected by directly eluting them from the HPLC column into the electrospray ion source of the mass spectrometer. The HPLC buffer underwent an electrospray ionization (ESI) of 1.6 kV using the liquid junction provided by the nanoelectrospray ion source, with the ion transfer tube temperature set to 200 °C. The LTQ-Orbitrap XL was operated in data-dependent mode by automatically switching between full-scan MS and consecutive MS/MS acquisitions. The full survey scan spectra (mass range of 400–2000) were acquired in the Orbitrap with a resolution of *r* = 60,000 at m/z 400, and the ion injection times for each spectrum were calculated to allow for the accumulation of 106 ions. In each survey, the ten most intense peptide ions (each having an intensity above 2000 counts and a charge state ≥ 2) were sequentially isolated at a target value of 10,000, then fragmented in the linear ion trap by collision-induced dissociation. Normalized collision energy was set to 35%, with an activation time of 30 milliseconds. To avoid selecting the same ion more than once, the peaks selected for fragmentation were automatically put on a dynamic exclusion list for 30 s, with a mass tolerance of ±10 ppm. The following parameters were used: the repeat count was set to 1; the exclusion list was limited to 500; singly-charged precursors were rejected; and the maximum injection time was set to 500 milliseconds for full-scan MS, or 300 milliseconds for MS/MS. The fragment ion spectra were recorded in the LTQ mass spectrometer in parallel with the Orbitrap full-scan detection. The Orbitrap was externally calibrated before each injection series to ensure an overall error rate under 5 ppm for the detected peptides. MS data were saved in the RAW file format using Xcalibur 2.0.7 and Tune 2.4 (Thermo Fisher Scientific, Bremen, Germany). A data analysis was performed using Proline 1.4 software, Mascot Distiller, and Mascot Server 2.5.1 (Matrix Science, Boston, MA, USA). The identification of peptides and proteins was done using the Mascot search engine via its automatic decoy database search, calculating a false discovery rate of 1% at the peptide level. MS/MS spectra were compared to the *Escherichia coli* Reference Proteome. For the MS and MS/MS experiments, mass tolerances were set to 10 ppm and 0.5 Da, respectively. The protein modifications used for this are as follows: fixed carbamidomethylation of cysteines, variable oxidation of methionine, variable acetylation of lysine, and variable acetylation of N-terminal residues.

#### 4.7.3. Cryo-Electron Microscopy

After diluting 150 nM ribosomes in buffer III (5 mM HEPES-KOH pH 7.5, 9 mM MgOAc, 50 mM KCl, 10 mM NH_4_Cl, and 1 mM DTT), 2 µL of each complex was applied to glow-discharged holey carbon grids (QUANTIFOIL R2/2), then flash-frozen in liquid ethane using a Vitrobot Mark III (FEI). The grids were transferred to a Cs-corrected Titan Krios electron microscope (FEI) operating at 300 kV. Using FEI’s EPU software 1.4.0, images were automatically recorded on their Falcon II direct electron detector set to a nominal magnification of 59,000× (corresponding to a calibrated pixel size of 1.16 Å), and a defocus range of −1.5 to −3.0 µm. The images were acquired in movie mode during exposures of 1 s (corresponding to 7 frames), under low-dose conditions (50 e^−^/Å^2^). Semi-automatic particle picking and reference-free 2D classification for discarding defective particles were both done using CryoSPARC 2.9 software [50], with 174,052 particles extracted from 922 micrographs. All subsequent steps used RELION 3.0.6 [51]. The original image stacks were corrected for drift and beam-induced motion, and summed using that software’s MotionCor2-like approach. For each micrograph, the contrast transfer function parameters were determined using CTFFIND 4.1.13 [52]. The initial 3D reconstruction was done using the 30S-portion of the crystal structure of the *E. coli* ribosome (PDB 4V4Q) [53] as the initial reference. This reference model was first low-pass filtered to a resolution of 40 Å to prevent bias toward high-frequency components in the map. The particles were then split into eight classes. As previously described [54,55], these classes represent several stages of 30S biogenesis, with the small subunit (SSU) body almost completely assembled, but the head at different levels of assembly. Unfortunately, the classes suffer from preferential-orientation artefacts. To further study the late stages of the 30S assembly, we combined the three classes that most resemble a mature 30S (subpopulation group IV). The resulting 48,873 particles were submitted to a second round of 3D refinement, followed by CTF refinement and polishing. A final round of 3D auto-refinement and post-processing produced a map with a global resolution of 5.9 Å with a Fourier shell correlation of 0.143. The local resolution map was estimated using Resmap 1.1.4 [56]. The consensus map was then used as a basis for multibody refinement [57] into two separate maps of the body and head of the SSU. The corresponding masks were made using a 30 Å low-pass filtered version of the consensus map, with soft edges of 12 pixels to define the solvent region boundary and ensure the overlapping of the two bodies. The resulting density maps were then sharpened using Phenix [58] and fitted on the consensus map using UCSF-Chimera [59]. To interpret the map, UCSF-Chimera was used to rigid-body fit the SSU-portion of the crystal structure of the *E. coli* (PDB 4YBB) mature 70S subunit [60] into the cryo-EM maps, with each protein and RNA treated separately. The atomic models of the protein and rRNA were then manually adjusted using COOT [61]. The entire model was further improved by real-space refinement against the consensus map using Phenix. The remaining analysis and the illustrations were undertaken using UCSF-Chimera.

## Figures and Tables

**Figure 1 ijms-24-03491-f001:**
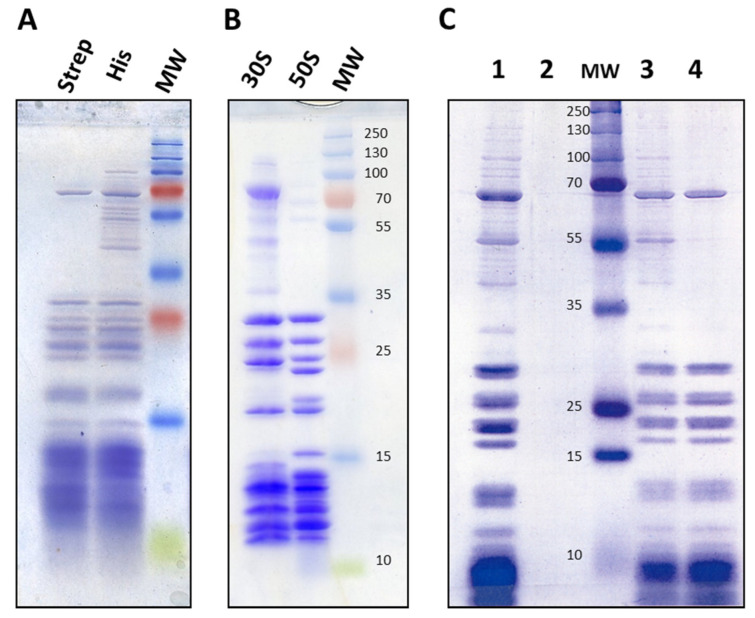
Ribosome purification and subsequent synthesis under heat stress. (**A**) Protein extracts from the Escherichia coli E6006 strain were either loaded onto streptactin columns and eluted with desthiobiotin (for Strep), or loaded onto NiTa resins and eluted with imidazole (for His). MW, molecular weight marker. (**B**) Purified ribosomes were loaded onto a streptactin resin, then flow-through (50S) and elution (30S) were analyzed. (**C**) MG1655 (pE7867) cells were grown overnight at 30 °C in LB medium containing 10 mM glucose, then inoculated in LB medium containing arabinose (lane 1) or glucose (2), and grown at 45 °C to an OD600 of 0.8. The same was done just in arabinose (1 mM), and grown at either 30 °C (3) or 45 °C (4). The ribosomes were loaded onto streptactin columns, and the eluted proteins were analyzed.

**Figure 2 ijms-24-03491-f002:**
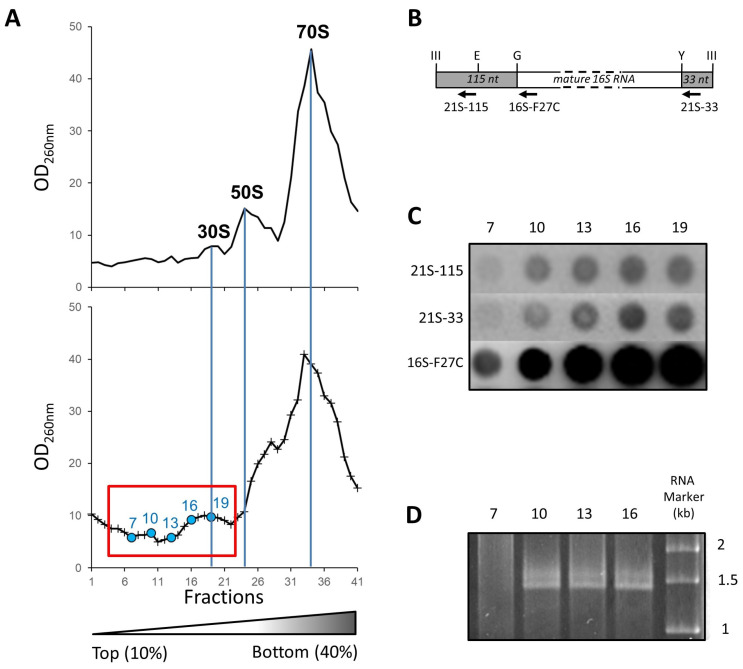
Identification of ribosomal precursors in sucrose-gradient fractions. (**A**) Sucrose density gradient profiles of purified ribosomal particles from the Escherichia coli E6006 strain before (top) and after (bottom) heat shock. The fractions 7 to 25 in 10–40% sucrose gradients (framed box) were used for purifying 21S ribosomal particles. The fractions with blue circles are those analyzed by slot blot hybridization (**B**,**C**) or RNA gel electrophoresis (panel D). (**B**) Schematic based on René and Alix [22] showing non-mature 16S rRNAs after cleavage by RNAse III. The cleavage generates the 115nt and 33nt extremities (ends). The ribonuclease cleavage sites are indicated: III for RNAse III; G for RNAse G; E for RNAse E; and Y for YbeY [27]. The biotinylated oligonucleotides used for hybridization against the dot blots (see panel C) are marked with arrows. (**C**) Hybridizations with the biotinylated oligonucleotides 21S-115, 21S-33, and 16S-F27C, target the non-mature SSU rRNA’s 115 nt and 33 nt ends as well as mature 16S rRNAs, respectively. The numbers on the top indicate the fractions shown after 2 µg of RNAs were applied on Amersham Hybond-N + nylon membranes. (**D**) Electrophoresis of RNA extracted from the fractions as numbered on top, with RNA sizing produced using Ambion Millenium RNA markers.

**Figure 3 ijms-24-03491-f003:**
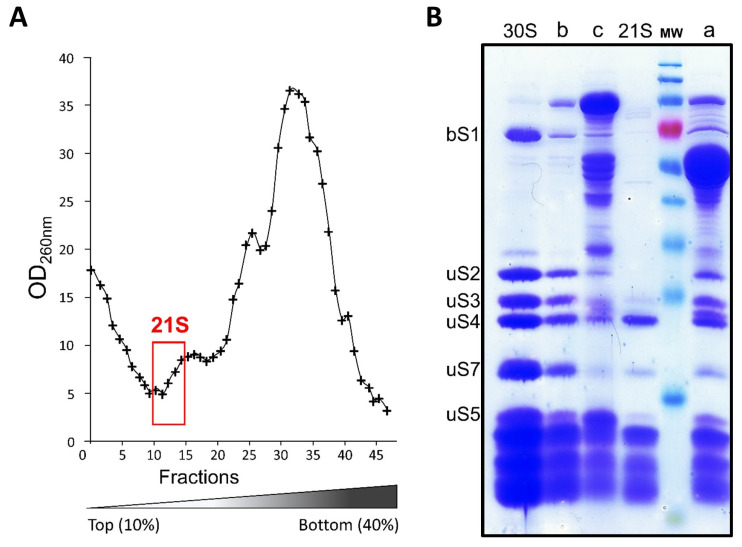
Purification of 21S particles. (**A**) *Escherichia coli* E6006 cells were grown in M63 medium and subjected to heat shock for 1 h at 45 °C. The clarified supernatants were loaded onto a 10–40% (*w*/*v*) linear sucrose gradient, and the sucrose density gradient profile is shown here. The fractions corresponding to 21S are outlined in red. (**B**) These fractions (“a”) were pooled, while 30S subunits were retained on NiTa resin and eluted with imidazole (“b”). The flow-through was applied to streptactin resin and eluted with desthiobiotin (21S), with the non-retained fraction (“c”) and the molecular weight (MW) markers shown. The bands corresponding to the ribosomal proteins bS1, uS2, uS3, uS4, uS5, and uS7 are also indicated.

**Figure 4 ijms-24-03491-f004:**
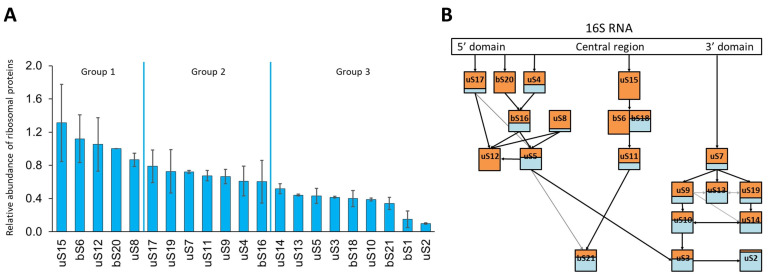
Relative quantification of 21S proteins by quantitative mass spectrometry. (**A**) Histogram of the ratio between purified 21S and 30S ribosomal particles analyzed by LC-MS/MS. Protein levels are normalized against those of the bS20 proteins used to purify the particles and are the means of three biological replicates, with standard deviation indicated. (**B**) Schematic representation of the proportion of ribosomal proteins in the Nomura assembly map, adapted from [7]. The ratio of proteins in 21S is orange, while the proteins from 30S are normalized to 1 and shown in blue.

**Figure 5 ijms-24-03491-f005:**
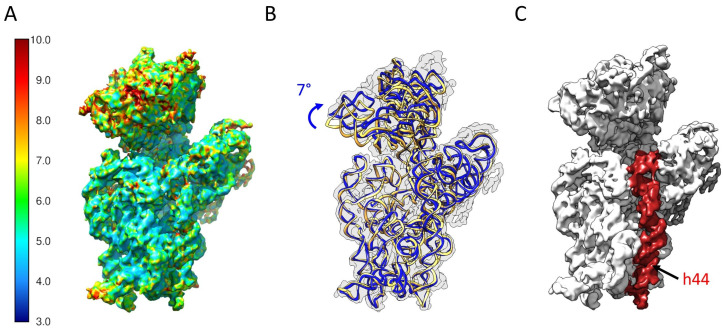
Cryo-EM structure of the 21S particle. (**A**) The map is colored to reflect the local resolutions computed with ResMap. (**B**) Comparison between the 16S rRNA fitted (blue) into the 21S map (light gray) and as in a mature 30S ribosomal subunit (yellow). The head of the 21S is rotated by 7° compared to the 30S. (**C**) The density corresponding to the helix h44 is highlighted in red.

**Figure 6 ijms-24-03491-f006:**
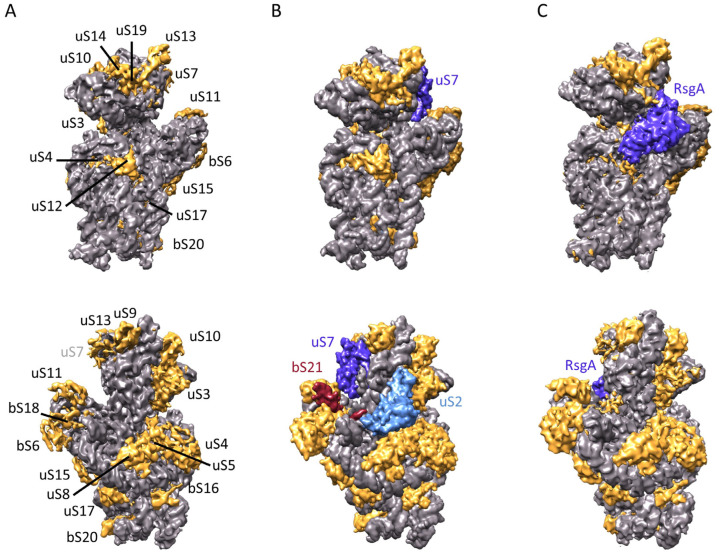
Structural comparison of 21S and 30S ribosomal subunits. (**A**) Front (top) and back (bottom) views of the 21S from this study, with the density corresponding to the rRNA in grey, and the ribosomal proteins highlighted in gold. Note that the density of uS7 is poorly defined. (**B**) Same views, but here of an atomic model of the *E. coli* 30S ribosomal subunit (PDB 4V4Q) filtered to adjust the resolution to match the 21S map. The ribosomal proteins uS2 and bS21, missing in our 21S structure, are light blue and red, while the uS7 is blue. (**C**) Same, but for the 30S + RsgA cryo-EM map (EMD-3663), with the RsgA maturation factor in blue.

**Table 1 ijms-24-03491-t001:** Strains and plasmids.

Strains		
MG1655	F^−^ λ^−^ *rph* 1	CGSC 6300 [48]
E5990	MG1655 *rpsB*:his-KanR	This study
E6001	MG1655 *rpsB*:his	This study
E5998	MG1655 *rpsT*:strep-kanR	This study
E6004	MG1655 *rpsT*:strep	This study
E6006	MG1655 *rpsB* his *rpsT* strep	This study
**Plasmids**		
pKD46	repA101(Ts) *araB*p*-gam-bet-exo* ori101 AmpR *araC*	[24]
pKD4	oriR6Kgamma AmpR *rgnB* KanR flanked by FRT	[24]
pCP20	Rep Ts CI857 AmpR CmR FLP	[24]
pBAD24	oriColE1 araC *araB*p	[49]
pE7867	pBAD24 rpsT:strep	This study

**Table 2 ijms-24-03491-t002:** Oligonucleotides used.

Type	Sequence	Experiment
rpsTREPF	GCACGTCATAAGGCTAACCTGACTGCACAGATCAACAAACTGGCTTGGAGCCACCCGCAGTTCGAAAAGTAAGTGTAGGCTGGAGCTGCTTC	rpsT strep construction
rpsTREPR	AACCCGCTTGCGCGGGCTTTTTCACAAAGCTTCAGCAAATTGGCGAATGGGAATTAGCCATGGTCC	
rpsBHISF	CAGGATCTGGCTTCCCAGGCGGAAGAAAGCTTCGTAGAAGCTGAGCACCACCACCACCACCACTAA GTGTAGGCTGGAGCTGCTTC	rpsB his construction
rpsFR	GCCTTTCTGCAACTCGAACTATTTTGGGGGAGTTATCAAGCCTTAATGGGAATTAGCCATGGTCC	
pkd4F	AAGCCATCCAGTTTACTTTGC	KanR insertion verification
pkd4D	CGCATCGCCTTCTATCGCC	
rpsBD	CGTTCTCAGGATCTGGCTTC	rpsT mutation Verification
rpsBF	CTCGGAGATGTGATCTGCC	
rpsTD	GTAAGCACAACGCAAGCCG	rpsT mutation Verification
rpsTF	ACAGAAGCCACTGGAGCAC	
rpsTA	TTTTCCATGGCTAATATCAAATCAGCTAAGAAGCGC	cloning of rpsT ORF
21S-115	5′ Biotin-TEG CATTTTTCGTGTTGCGACG	detection of 17S
21S-33	5′ Biotin-TEG CAAAGAACGCTTCTTTAAG	
16S-F27C	5′ Biotin-TEG CTGAGCCATGATCAAACTCT	

## Data Availability

The maps were deposited with the EMDB under the accession code EMD-16556.

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
