# Peer review of "Purification and Characterization of Authentic 30S Ribosomal Precursors Induced by Heat Shock"

_ijms, 2023, doi:10.3390/ijms24043491_

Round 1

Reviewer 1 Report

The ribosome is a large macromolecular complex of 57 individual components. Ribosomes can be subdivided into two subunits – small and large ribosomal subunits. To be assembled, each of these subunits undergoes a complex process of many steps, such as rRNA transcription, processing and modification; ribosomal proteins synthesis, and final binding of all components to each other.

In the present manuscript, Guidice et al., studied the late assembly intermediate of the small ribosomal subunit. The authors took an elegant approach and used heat shock as the tool to stop the ribosome assembly in the late stage in vivo. The authors optimized the late assembly intermediates purification method based on the sucrose gradient density centrifugation and affinity purification combination. It allowed the collecting of enough ribosome assembly intermediates for their characterization by quantitative mass-spectrometry and cryo-EM structure determination methods. Authors showed that late-assembly ribosomal proteins such as uS2, uS3, uS7, and bS21 are predominantly absent in the late ribosomal intermediates. Therefore, the approach presented here can also be used to study other ribosomal intermediates.

This manuscript will be interesting for the broad auditorium. Experimental results support all conclusions. The manuscript is well-written, and all previous essential works are cited.  Therefore, I recommend it for publication in its present form.

Author Response

We deeply thank reviewer 1 for its encouraging remarks.

Reviewer 2 Report

The study by Giudice and colleagues uses an impressive array of approaches to characterise 30S ribosome precursors that accumulate upon heat shock. However, presentation of the results requires additional work, both the text and the figures. For instance, the introduction is written as a wall of text. Please, split into paragraphs. The figures are sloppy, there is no common visual language. Please put some effort into making the paper look presentable.

Specific comments:

p. 1, line 39: ‘They are also central actors as they sense the cell’s metabolic state 39 through their interactions with RelA.’ It is not just RelA; it is both RelA and Rel. For RSH family one can cite PMID: 21858139. Also: ‘(p)ppGpp’, not ‘ppGPP’.

p. 1, line 41: Please cite PMID: 32176689 as a reference for Rel being involved in heat shock.

Results: was the functionality of tagged proteins tested? Could one do a sucrose gradient supplemented with Western?

Figures in general are aesthetically not very pleasing. They are not formatted in a standard way (compare scatter plots on Figure 2A vs Figure 3A).

Figure 1: gels are boxed in. Figure 2 – not. Please standartise.

Figure 2A: something is seriously wrong with recognition of symbols, they are garbled with question marks all over the place. Please add the gradient trace (sucrose %), mark top and bottom.

Figure 3A: the y axis is not labelled. Please standardise with Figure 2A, add the gradient trace (sucrose %), mark top and bottom. 3B: something is up with the top of the figure.

Line 237: CryoSPARC, not cryosSparc

Line 234: italicise ΔrimM

Author Response

We thank reviewer 2 for its insightful comments. They were all addressed in the text and described below:

The study by Giudice and colleagues uses an impressive array of approaches to characterise 30S ribosome precursors that accumulate upon heat shock. However, presentation of the results requires additional work, both the text and the figures. For instance, the introduction is written as a wall of text. Please, split into paragraphs. The figures are sloppy, there is no common visual language. Please put some effort into making the paper look presentable.

--> Thank you for the general comment. The introduction was indeed carelessly edited and has now been improved. The figures were reshaped as suggested, making the article much more presentable:

Specific comments:

1, line 39: ‘They are also central actors as they sense the cell’s metabolic state 39 through their interactions with RelA.’ It is not just RelA; it is both RelA and Rel. For RSH family one can cite PMID: 21858139. Also: ‘(p)ppGpp’, not ‘ppGPP’.

--> Rel and RSH roles were added in the intoduction paragraph, with the suggested references

1, line 41: Please cite PMID: 32176689 as a reference for Rel being involved in heat shock.

--> Done

Results: was the functionality of tagged proteins tested? Could one do a sucrose gradient supplemented with Western?

--> We confirmed the correct activity of the S2-His tagged ribosomes by in vitro translation assays and they are indeed functional (not shown). Moreover the proteins are clearly present in the MS analysis, so a WB would only confirm that.

Figures in general are aesthetically not very pleasing. They are not formatted in a standard way (compare scatter plots on Figure 2A vs Figure 3A).

--> Figures 2 and 3 were improved, according to reviewer's 2 remarks

Figure 1: gels are boxed in. Figure 2 – not. Please standartise.

--> done

Figure 2A: something is seriously wrong with recognition of symbols, they are garbled with question marks all over the place. Please add the gradient trace (sucrose %), mark top and bottom.

--> fixed

Figure 3A: the y axis is not labelled. Please standardise with Figure 2A, add the gradient trace (sucrose %), mark top and bottom. 3B: something is up with the top of the figure.

--> Fixed

Line 237: CryoSPARC, not cryosSparc

--> Done

Line 234: italicise ΔrimM

--> Done

Round 2

Reviewer 2 Report

The authors revised the ms sufficiently.

Author Response

Thanks again for your insightful comments